Management of investment portfolios employing reinforcement learning

Santos Gustavo Carvalho 1
Garruti Daniel 2
Barboza Flavio flmbarboza@ufu.br 2
de Souza Kamyr Gomes 2
Domingos Jean Carlos 2
Veiga Antônio 1
1 Faculdade de Engenharia Elétrica, Universidade Federal de Uberlândia , Uberlândia , MG , Brazil
2 Faculdade de Gestão e Negócios, Universidade Federal de Uberlândia , Uberlândia , MG , Brazil
Schaerf Andrea
Electronic publication date: 2023 Dec 11
Publication date: 2023
Volume: 9
Electronic Location ID: e1695
Received 2023 Aug 7; Accepted 2023 Oct 23
Copyright: ©2023 Santos et al.
Copyright year: 2023
Copyright holder: Santos et al.
License: This is an open access article distributed under the terms of the Creative Commons Attribution License, which permits unrestricted use, distribution, reproduction and adaptation in any medium and for any purpose provided that it is properly attributed. For attribution, the original author(s), title, publication source (PeerJ Computer Science) and either DOI or URL of the article must be cited.
License URL: https://creativecommons.org/licenses/by/4.0/

Keywords: Reinforcement learning, Finance, Portfolio optimization, Investment, Stock market, Data-driven investing, Market risk management

Funding: Sapiens Agro (Sapiens Inteligência Ltda) The authors received no funding for this work. Sapiens Agro (Sapiens Inteligência Ltda) provided funding for the APC. The funders had no role in study design, data collection and analysis, decision to publish, or preparation of the manuscript.

==============================
Investors are presented with a multitude of options and markets for pursuing higher returns, a task that often proves complex and challenging. This study examines the effectiveness of reinforcement learning (RL) algorithms in optimizing investment portfolios, comparing their performance with traditional strategies and benchmarking against American and Brazilian indices. Additionally, it was explore the impact of incorporating commodity derivatives into portfolios and the associated transaction costs. The results indicate that the inclusion of derivatives can significantly enhance portfolio performance while reducing volatility, presenting an attractive opportunity for investors. RL techniques also demonstrate superior effectiveness in portfolio optimization, resulting in an average increase of 12% in returns without a commensurate increase in risk. Consequently, this research makes a substantial contribution to the field of finance. It not only sheds light on the application of RL but also provides valuable insights for academia. Furthermore, it challenges conventional notions of market efficiency and modern portfolio theory, offering practical implications. It suggests that data-driven investment management holds the potential to enhance efficiency, mitigate conflicts of interest, and reduce biased decision-making, thereby transforming the landscape of financial investment.

Introduction

The financial markets display a high degree of dynamism and complexity, making the selection of an optimal combination of assets for constructing an investment portfolio a formidable challenge (Song et al., 2022; Xiao & Ihnaini, 2023 among others). In this context, scholars have conducted thorough investigations into Modern Portfolio Theory (MPT) of Markowitz (1952) since its inception. With the advent of technological advancements, researchers have increasingly delved into advanced artificial intelligence (AI) models, particularly within the subfield of machine learning, such as Reinforcement Learning, to augment investment management and introduce innovative investment strategies.

The optimization of investment portfolios has been a subject of active discussion in the field of Finance. This discussion arises from the necessity to diversify assets for risk mitigation and return maximization. Notable studies by Rubinstein (2002); Wilford (2012), and Millea & Edalat (2022) underscore the relevance of Modern Portfolio Theory (MPT). However, the examination of reinforcement learning has gained momentum in response to the challenges associated with adhering to MPT’s premises. These challenges have raised questions about the feasibility of achieving ideal diversification, particularly concerning the rationality of market participants.

The theoretical framework proposed by Lo (2004) seeks to address this exigency by questioning market efficiency and supporting the idea of markets adapting to novel scenarios, termed the Adaptive Markets Hypothesis (AMH). This perspective ushers in the potential employment of techniques such as RL. Millea & Edalat (2022) and Lin & Beling (2020) evince the applicability of RL within the investment domain, further demonstrating the superiority of RL over MPT in managing portfolios.

This research endeavor seeks to assess the effectiveness of reinforcement learning algorithms in optimizing investment portfolios. Utilizing the FinRL library and five specific algorithms outlined in subsequent sections, the study conducts a comparative analysis of the results obtained through these algorithms in contrast to conventional strategies. These traditional approaches include Minimum Variance (MINVAR), as proposed by Markowitz (1952), and the Buy-and-Hold (B&H) strategy.

In addition, this study includes the Dow Jones and Ibovespa indices as benchmark references for calculating beta and alpha metrics, with the aim of assessing the performance and risk of optimized portfolios. The primary objective is to evaluate the effectiveness of reinforcement learning algorithms in comparison to conventional strategies and to gain insights into the implications of incorporating commodity derivatives into an investment portfolio.

Another significant aspect explored in this study involves the examination of the influence of transaction costs on model performance. The research investigates how models adapt to environments both with and without transaction costs, shedding light on the adaptability of algorithms to the dynamic realities of financial markets.

This article aims to advance the field of finance by providing insights into the application of machine learning techniques for optimizing investment portfolios. It emphasizes the significance of diversification through the inclusion of derivatives and investigates the impact of transaction costs, contributing to a deeper understanding of the opportunities and challenges associated with these approaches.

It is imperative to highlight that the outcomes vary when considering transaction costs. For instance, when accounting for these costs, returns can even be greater than when not considered, which could be attributable to the algorithm abstaining from executing certain transactions or selecting alternative assets. Such findings concur with Millea & Edalat (2022) and evince satisfactory returns in accordance with Fama (1965).

This investigation contributes to scientific literature and practice in the domain of Finance in several ways. Primarily, it proffers exemplars of the application of advanced machine learning techniques, such as RL, in investment portfolio optimisation. By juxtaposing the performance of these techniques with traditional strategies, such as MINVAR and B&H, the study posits that AI techniques can be more efficacious in maximising investment returns without the necessity for presuppositions mandated by traditional theory, whilst also demanding less investor engagement in analyses and portfolio rebalancing.

Moreover, this study underscores the importance of diversifying assets, including the incorporation of commodities as a viable alternative. It showcases that investments in these derivatives have the potential to significantly enhance portfolio performance while concurrently mitigating volatility. Such findings hold paramount importance for investors aiming to enhance the profitability of their investments while effectively managing portfolio risk.

Another significant contribution of this study is the examination of the influence of transaction costs on RL model training. Through an exploration of how models adapt to environments both with and without transaction costs, this study illuminates the adaptability of algorithms to diverse financial market conditions. This analysis holds paramount importance in gaining insights into the possibilities, limitations, and challenges associated with the utilization of AI techniques in investment portfolio management.

In conclusion, this research offers pertinent implications for the domain of Finance. Firstly, the findings suggest that employing RL can contribute to enhancing risk management and augmenting investment returns. Additionally, the study accentuates the importance of asset diversification in portfolios, particularly through the inclusion of commodities derivatives.

Secondly, alternative investments hold appeal for investors aiming to further enhance their performance, utilizing diversification to achieve superior results while maintaining lower portfolio risk levels. Another noteworthy implication is the substantial influence of transaction costs. This underscores the importance of a thoughtful consideration of transaction costs during model development and evaluation, enhancing the realism and accuracy of results.

The structure of this article follows a systematic organization. The next section introduces the theoretical framework, while the Methods section presents the data and the proposed methodology. The Results section provides findings and a discussion in the context of existing literature. Finally, the last section offers the study’s conclusion, including insights into limitations and potential avenues for future research.

Theoretical Aspects

Portfolio management

The theoretical underpinning of portfolio management is extensive, with several key facets deserving recognition. Modern Portfolio Theory (MPT), as introduced by Markowitz (1952), represents a pivotal perspective that has received substantial attention in both academic discourse and among practitioners in the investment arena.

In essence, MPT, as articulated by Markowitz (1952), underscores the opportunity for investors to enhance returns while mitigating risk through strategic diversification. This strategy is rooted in an analysis of historical asset performance and volatility. Subsequently, a mathematical formulation, underpinned by an optimization model, calculates the allocation of each asset within the portfolio. This optimization aims to maximize expected returns for a given level of risk or, conversely, minimize risk. This approach leads to the identification of the most effective portfolio structure likely to yield optimal returns under specific circumstances, often referred to as the minimum variance portfolio (Millea & Edalat, 2022). It is important to note that a higher number of assets in a portfolio doesn’t necessarily equate to prudent diversification. For instance, assets may be concentrated within a single sector, potentially amplifying returns while exposing the portfolio to equivalent risk levels.

Evolutions enveloping the theory emanate across diverse fronts, principally attributable to the challenges associated with complying with the underlying assumptions, which, if unmet, could vitiate any analysis or critique of MPT. Notably, the assumptions of market participant rationality and market efficiency represent characteristics that are incessantly challenged within the literature (Wilford, 2012).

The Efficient Market Hypothesis (EMH) postulates that asset prices in financial markets follow random and independent movements, making it impervious to prediction, even when historical data is analyzed using technical analysis. This is due to the inherent uncertainty of news, which is widely and instantaneously accessible (Fama, 1965). Consequently, the current price is believed to accurately reflect the intrinsic value of the asset, making fundamental analysis unnecessary. The hypothesis asserts that the most reasonable expectation for the future price is the current price, and any return above the market average is considered exceptional.

In contrast, the Adaptive Market Hypothesis (AMH) posits that financial markets undergo cyclical fluctuations between states of efficiency and inefficiency, which are influenced by external factors such as geopolitical conflicts and political interventions (Lo, 2004). This hypothesis further contends that investors display bounded rationality and demonstrate discernible behavioral patterns.

An increase in research efforts directed at formulating investment portfolios has been observed, driven by the proliferation of accessible data and the introduction of innovative methodologies. In light of these advancements, a comprehensive survey by Loke et al. (2023) delineates the developments in the Portfolio Optimization Problem (POP) from 2018 to 2022. The paper categorizes contemporary solution techniques, highlighting key areas, including metaheuristics, mathematical optimization, hybrid approaches, matheuristics, and machine learning.

Significantly, the survey highlights a growing interest in hybrid methodologies, particularly noticeable since 2018. The findings presented by Loke et al. (2023) emphasize the importance of acknowledging and addressing the emerging trends and gaps in this field. This expansion has resulted in noteworthy improvements in the outcomes obtained (Jang & Seong, 2023).

However, there is room for additional exploration within this field, particularly by leveraging artificial intelligence (AI) techniques such as deep learning and reinforcement learning (RL), which have garnered increasing attention in recent research. Furthermore, there are opportunities to integrate non-traditional assets, including cryptocurrencies, commodities, and indices, into such research endeavors (Santos et al., 2022).

Reinforcement learning

Reinforcement learning (RL) represents a subfield within machine learning, with a primary focus on sequential decision-making in uncertain and stochastic environments. The central objective of RL is to determine optimal policies that maximize cumulative rewards over time (Sutton & Barto, 2018). An RL problem comprises fundamental elements, including an agent, environment, states, actions, and rewards. The agent interacts with the environment by executing actions based on its current state, thereby receiving rewards in return, all while endeavoring to acquire a policy that maximizes the accumulated reward.

An essential element in the realm of reinforcement learning is the state-value function, represented as V(s), which quantifies the anticipated value of future rewards accumulated from state s under a specific policy π (Sutton & Barto, 2018). Furthermore, Bellman’s equation, a recursive relationship, establishes a linkage between the current state’s value and the values of subsequent states, thereby facilitating state-value updates and streamlining the pursuit of the optimal policy (Sutton & Barto, 2018). Bellman’s equation for the state-value function is expressed as follows: (1) Vs= ∑aπa|s∑s′,rps′,r|s,ar+γVs′,

wherein p(s′, r|s, a) denotes the environment transition function, π(a|s) represents the agent’s policy, and γ is the discount factor.

Deep reinforcement learning (DRL) integrates reinforcement learning with deep neural networks, thereby enabling the acquisition of optimal policies in scenarios characterized by high-dimensional state and action spaces (Mnih et al., 2015). DRL has garnered remarkable achievements across a broad spectrum of complex tasks, including gaming, robotics, and system optimization (Silver et al., 2017; Gu et al., 2017; Mnih et al., 2015). This advancement distinguishes itself from conventional RL by its capacity to tackle challenges of greater scale and complexity, which would otherwise encounter computational and representational limitations within the classical RL domain.

The Q-table comprises a matrix that preserves the value associated with each state-action pair, denoted as Q(s, a), where ‘value’ signifies the expected reward upon executing action a in state s and subsequently adhering to policy π (Sutton & Barto, 2018). Conversely, Deep Q-Network (DQN) represents an approach that amalgamates the Q-table concept with deep neural networks, replacing the Q-table with a neural network that approximates the state-action value function, known as the Q-function (Mnih et al., 2015). This modification empowers the algorithm to effectively manage substantially larger and more intricate state and action spaces, generalizing the Q-function while providing superior scalability and efficiency compared to the traditional Q-table. Bellman’s equation for the Q-function is articulated as: (2) Qs,a= ∑s′,rps′,r|s,ar+γmaxa′Qs′,a′,

wherein: Q(s, a) represents the Q-function that depicts the expected value of accumulated future rewards when action a is taken in state s followed by the adherence to the optimal policy. s′ denotes the subsequent state in the environment after executing action a in state s. r indicates the immediate reward procured after performing action a in state s. p(s′, r|s, a) represents the environment’s probability transition function, describing the transition probability to the subsequent state s′ and receiving reward r upon taking action a in state s. γ is the discount factor, ranging between 0 and 1, dictating the relative significance of future rewards compared to immediate rewards. Values close to 0 lead the agent to value immediate rewards predominantly, whilst values approaching 1 make the agent consider future rewards in a more balanced manner. maxa′Q(s′, a′) expresses the maximal value of the Q-function for the subsequent state s′, considering all potential actions a′.

Bellman’s equation for the Q-function facilitates the iterative update of Q-function values, refining the estimates of expected values pertaining to future rewards accumulation, thus contributing to the pursuit of the optimal policy.

Unlike traditional machine learning, which predominantly focuses on supervised or unsupervised learning paradigms, reinforcement learning (RL) techniques are dedicated to sequential learning and decision-making (Goodfellow, Bengio & Courville, 2016). One notable advantage of these methodologies lies in their ability to acquire knowledge directly through interactions with the environment, eliminating the reliance on labeled data. However, it’s important to note that RL techniques can demand substantial computational resources and extended training periods when compared to classical machine learning methods. Additionally, RL and deep reinforcement learning (DRL) problems can exhibit sensitivity to problem formulation, including the specification of rewards and states, necessitating careful adjustments and empirical exploration.

To comprehend the state-of-the-art concerning the application of RL and its variations within portfolio management, Santos et al. (2022) conducted an exhaustive literature review on the seminal works related to the application of artificial intelligence in portfolio management. This systematic literature review renders a comprehensive perspective on the advancements and challenges in the domain, underscoring studies that employ RL and its variants, thereby enabling a more profound understanding of the potential and limitations of these techniques in investment portfolio management.

Literature pertaining to portfolio management via RL

Thanks to technological advancements, the field of finance is subjected to a more comprehensive examination, considering the myriad of available investment strategies. The introduction of RL algorithms has prompted academic literature to reevaluate the domain of investment portfolio management, with a focus on their application and exploration within the financial context. This section highlights some of the key foundational works.

Jang & Seong (2023) employed a deep reinforcement learning technique to enhance the formation of equity portfolios of the S&P 500 index, employing a neural network as the learning agent. The methodology takes into account technical analysis indicators and market trend information to substantiate decisions regarding resource allocation in equities. According to the authors, this approach excels in comparison to other techniques, as it considers asset allocation in accordance with the market trend. The findings suggest that this technique holds promise as an alternative for portfolio optimisation in complex and dynamic financial markets.

In the study steered by Millea & Edalat (2022), DRL found its utility in optimising portfolios that included equities, currency pairs from the Forex market, and cryptocurrencies—a vanguard in relation to prior studies. The authors amalgamated DRL models and hierarchical clustering into a decision-making system to assign weights to an asset portfolio. The DRL agent acquired the proficiency to alternate between low-level models, culminating in superior performance compared to individual models or a random policy. The outcomes indicated that portfolios inclusive of cryptocurrencies exhibited a superior performance in terms of risk-adjusted returns. Nonetheless, it is imperative to accentuate that the cryptocurrency market is highly volatile and harbours significant risks. Thus, the inclusion of these assets in portfolios mandates circumspection and meticulous analysis. Moreover, the authors bolster the notion that ample real-world evidence indicates market efficiency’s shortcomings at various junctures, which implies an ill-suited environment for the application of MPT, consequently substantiating the utility of the adaptations proffered by DRL models.

Song et al. (2022) introduced an innovative method for optimizing investment portfolios utilizing stochastic reinforcement learning. To validate their model, the authors conducted an empirical analysis using data from 22 stocks with the highest trading volume in the S&P 500 index from 2005 to 2020. The model’s performance was evaluated both before and during the COVID-19 crisis. The findings demonstrated that the proposed approach outperformed the benchmark, traditional stochastic models, and popular algorithms, achieving higher returns while maintaining lower risk.

Methods

The methodology employed in this study leverages reinforcement learning (RL) techniques using the open-source FinRL library (Liu et al., 2020). The portfolio management involves a diverse set of assets, including 40 equities equally distributed across Brazilian, American, European, and Chinese markets, in addition to futures prices of 10 commodities (rice (ZR), live cattle (LE), coffee (KC), ethanol(FL), corn (ZC), iron ore (TR), gold (GC), crude oil (CB), soybeans (ZS), and wheat (ZW)). Historical equity data was sourced from Yahoo Finance (2022), a common data provider in similar studies (Xiao & Ihnaini, 2023), while commodity data was obtained from Barchart (2022). The selection of equities for each country’s portfolio was based on the ranking of the most traded stocks available on Investing.com (2022).

The assemblage of assets, including commodities, aims to ensure appropriate portfolio diversification and to reflect the significance of the agricultural sector in Brazil. As articulated by Markowitz (1952), diversification continues to be a fundamental strategy for risk mitigation and portfolio enhancement. Moreover, commodities play an indispensable role in the Brazilian economy, with the nation ranking amongst the world’s largest exporters of agricultural products (Pereira et al., 2012).

The FinRL library proffers an extensive assortment of reinforcement learning algorithms, along with tools for training evaluation and result analysis. This study endeavours to probe the deployment of reinforcement learning techniques for asset selection and the management of a diversified portfolio, with the aspiration of outperforming conventional investment strategies. To this end, historical price data for stocks and commodities, as well as technical indicators, were employed to train and evaluate reinforcement learning models across various market scenarios, such as bullish and bearish periods, pandemics, and economic crises. These scenarios contribute to understanding how models acclimate to market fluctuations and the challenges posed by global events and adverse economic conditions.

Four distinct combinations of investment portfolios underwent testing, with two of them incorporating commodities and the other two excluding them. Similarly, two portfolios were evaluated: one considering a standard 1% transaction fee, as commonly used in the FinRL library (Liu et al., 2020), and the other without such a fee. This allowed for an assessment of the impact of transaction costs on risk metrics, including the Sharpe ratio and beta. Furthermore, the study compared the outcomes generated by the RL technique with those of the MINVAR and B&H strategies, using the Dow Jones and Ibovespa indices as benchmarks.

Comparisons with the MINVAR and B&H strategies were executed to evaluate the efficacy of reinforcement learning techniques relative to more traditional investment approaches (Yang, Liu & Wu, 2018). This comparison is instrumental in determining whether reinforcement learning models can offer noteworthy advantages in terms of performance and risk management. The significance of the Dow Jones and Ibovespa is highlighted in the work of Vartanian (2012), where the author scrutinises the relationship between stock and bond returns and common risk factors, employing the Dow Jones as a representative benchmark of the US stock market. Moreover Vartanian (2012), examines the literature on systemic financial risk and underscores the importance of Ibovespa in analysing Brazil’s economic and financial performance, examining the impacts of the Dow Jones index, commodities, and exchange rates on Ibovespa and the contagion effect amongst these markets.

To assess model performance, the dataset was divided into training and test sets. The training set comprised data collected from January 4, 2013, to January 1, 2018, while the test set included data gathered from January 2, 2018, to October 27, 2022. During the training phase, algorithms underwent training to optimize their decision-making and maximize rewards within the given environment. Subsequently, model performance was evaluated using the test set, where models were required to make decisions based on data not previously encountered during training.

The allocation of approximately 53.64% of the data to the training set and 46.36% to the test set aimed to achieve a balanced dataset and mitigate the risk of overfitting, considering the temporal nature of the data. This approach aligns with common practices in artificial intelligence and machine learning, as demonstrated in seminal works such as Cawley & Talbot (2010) on overfitting and model selection, as well as Hyndman & Athanasopoulos (2018) on time series forecasting, where the train-test split is a crucial component for assessing model generalization.

The choice of a specific cut-off date between data collection periods ensures that the model undergoes training on historical data and subsequent testing on more recent data, thereby providing a better representation of real-world scenarios. This method aligns with established practices in time series and forecasting research, as demonstrated in prior studies such as Hochreiter & Schmidhuber (1997) and Xiao & Ihnaini (2023), which employ Long Short-Term Memory (LSTM) models, as well as Santos et al. (2021), who used LSTM for forecasting Brazilian ethanol spot prices. In these works, the partitioning of data into training and testing sets is guided by the temporal nature of the data, facilitating the model’s ability to generalize to temporal changes in the dataset.

In this study, it was conducted an analysis and comparison of results obtained from five distinct reinforcement learning algorithms: Advantage Actor-Critic (A2C), Deep Deterministic Policy Gradient (DDPG), Proximal Policy Optimization (PPO), Soft Actor-Critic (SAC), and Twin Delayed Deep Deterministic Policy Gradient (TD3). The selection of these algorithms was based on their representativeness within the field of reinforcement learning, encompassing key approaches and advancements. Furthermore, the availability of these algorithms within the FinRL library facilitated their direct comparison and comprehensive analysis.

As highlighted in Liu et al. (2021), reinforcement learning algorithms can be categorized into value-based, policy-based, and actor-critic types. Q-learning Watkins & Dayan (1992) is a value-based method that employs a Q-table to address problems with small state and action spaces. Advanced techniques like DQN and its variants Achiam (2018) utilize deep neural networks to handle more complex spaces. On the other hand, policy-based algorithms Sutton et al. (2000) directly adjust the policy parameters using a gradient approach, bypassing the need for value estimation. Actor-critic algorithms combine the advantages of both value-based and policy-based methods by updating two neural networks: the actor network, which updates the policy, and the critic network, which estimates the state-action value function. During training, the actor network takes actions that are subsequently evaluated by the critic network. All algorithms utilized in this work are of the actor-critic type Achiam (2018).

The Advantage Actor-Critic (A2C) algorithm (Mnih et al., 2016) utilizes a sample-based approach, employing multiple agents to concurrently update both the policy and the value function. This method extends the Actor-Critic framework (Rosenstein et al., 2004) by integrating a policy model (the actor) with a value function (the critic) to facilitate optimal policy learning. A2C offers several advantages, including its ability to handle continuous action spaces and high-dimensional observations, as well as its capacity to learn scalable and stable policies in multi-agent training settings (Wang et al., 2016). However, it does come with limitations, such as the potential to converge to a local minimum instead of a global optimum and the challenge of appropriately tuning hyperparameters (Sutton & Barto, 2018). Nevertheless, A2C remains one of the most popular and effective reinforcement learning algorithms available, with numerous applications in gaming, robotics, and various other domains (Vinyals et al., 2019; Dhariwal et al., 2017).

The Deep Deterministic Policy Gradient (DDPG) algorithm, as introduced by Lillicrap et al. (2015), extends the Deterministic Policy Gradient (DPG) algorithm with the aim of learning deterministic policies for continuous problems. DDPG employs an off-policy learning approach, enabling the utilization of past experiences to enhance efficiency. Furthermore, DDPG relies on deep neural networks to approximate both the value function and the policy. Its primary limitations include sensitivity to hyperparameters and training instability (Henderson et al., 2018).

Proximal Policy Optimization (PPO), as elucidated by Schulman et al. (2017), offers stable and efficient training in contrast to traditional Policy Gradient methodologies. By employing a clipping surrogate objective function, PPO adeptly balances exploration and exploitation, effectively mitigating the risk of suboptimal policy adjustments. Its versatility is underscored by its successful applications across diverse domains, including robotics Heess et al. (2017), gaming (notably achieving exceptional performance in Dota 2) Berner et al. (2019), finance Lin & Beling (2020), and wind farm management Pinciroli et al. (2021).

Soft Actor-Critic (SAC), as introduced by Haarnoja et al. (2018), represents an off-policy algorithm that combines actor-critic techniques with maximum entropy optimization. This amalgamation achieves a finely tuned equilibrium between exploration and exploitation. SAC stands out in diverse continuous control tasks, particularly in the realm of robotics for intricate undertakings like object manipulation, as demonstrated in Haarnoja et al. (2019), and in control simulations for optimizing autonomous vehicles, as shown in Zhao et al. (2020).

The Twin Delayed Deep Deterministic Policy Gradient (TD3), as addressed by Fujimoto, Hoof & Meger (2018), represents an enhancement of the Deep Deterministic Policy Gradient (DDPG) algorithm, achieved through the incorporation of delayed actor updates, dual critic networks, and targeted action noise. These refinements contribute to heightened stability when dealing with continuous control problems. TD3 has demonstrated its efficacy in various applications, including robotics for object manipulation Veeriah, Venkatraman & Goldberg (2020) and control simulations aimed at optimizing traffic for autonomous vehicles, as shown in Aboudolas & Roussaki (2020).

In the context of hyperparameters, this study strictly adheres to the guidelines provided by the FinRL library and replicates the same hyperparameters as specified in a tutorial offered by the FinRL team (AI4Finance-Foundation, 2021). This choice is made to facilitate the replication of the research. FinRL takes charge of initializing the agent class within the provided environment, deploying the Deep Reinforcement Learning (DRL) algorithm with the aforementioned hyperparameters (model_kwargs), and overseeing the training regimen to yield a trained model. This procedural overview is captured in Table 1. It’s worth noting that the impact of varying hyperparameters could be a fruitful avenue for future research, particularly for studies focused on understanding how different algorithms respond to changes in these settings.

Table 1 Functions for creating and training DRL agents (Liu et al., 2021).

Function	Description	
env = StockTradingEnv (df, **env_kwargs)	Returns an instance of the Env class with data and default parameters.	
agent = DRLAgent(env)	Instantiates a DRL agent with a given environment env.	
model = agent.get_model(model_name, **model_kwargs)	Returns a model with a specified name and default hyperparameters.	
trained_model = agent.train_model (model)	Initiates the training process for the agent and returns a trained model.	
Notes.

Source: Adapted from Liu et al. (2021).

The reward function plays a pivotal role in the design and implementation of reinforcement learning algorithms, as it establishes the objective that the agent must pursue throughout the learning process. In this study, the cumulative return has been selected as the reward function, in line with a widely adopted approach in related literature. The rationale for this choice stems from the fact that cumulative return enables effective evaluation of the agent’s performance in terms of long-term outcomes, fostering the development of more robust and efficient policies.

In terms of hyperparameters, this study adopts the default values provided by the FinRL library for each algorithm. This choice is made to maintain a fair and consistent basis for comparing the various methods under analysis. The utilization of default hyperparameters facilitates the evaluation of each algorithm’s performance under uniform conditions, allowing for a more precise analysis of their capabilities and limitations within the context of the problem being investigated.

The state space of the models is formulated using a range of inputs that are relevant to the analysis of financial assets. In a manner akin to the approach presented by Santos et al. (2021), asset returns over 20, 40, and 60 days are incorporated. Additionally, the technical indicators utilized include the Relative Strength Index (RSI), Stochastic Oscillator, Williams %R (WILLR), Moving Average Convergence Divergence (MACD), Rate of Change (ROC), and On Balance Volume (OBV). The covariance matrix is also employed in the construction of the state space, following the methodology outlined by Liu et al. (2020).

The technical indicators used are defined as follows, as presented by Murphy (1999):

• RSI (Relative Strength Index): (3) RSI=100−1001+RS

Wherein: RS=Average gain over the last 14 daysAverage loss over the last 14 days.

• Stochastic Oscillator: (4) %K=100∗C−L14H14−L14

Wherein: C represents the current closing price; L14 denotes the lowest price in the last 14 days; H14 signifies the highest price in the last 14 days.

• MACD (Moving Average Convergence Divergence): (5) MACD=EMA12C−EMA26C

(6) Signal Line=EMA9MACD

Wherein: C denotes the time series of closing data; EMAn signifies the n-day exponential moving average; H14 represents the highest price in the last 14 days.

• ROC (Rate of Change): (7) PROCt=Ct−Ct−nCt−n

Wherein: PROC(t) denotes the rate of change of the price at time t; C(t) represents the closing price at time t.

• OBV (On Balance Volume): (8) OBVt=OBVt−1+Volt,ifCt>Ct−1OBVt−1−Volt,ifCt<Ct−1OBVt−1,ifCt=Ct−1

Wherein: OBV(t) denotes the on balance volume indicator at time t; Vol(t) signifies the trading volume at time t; C(t) represents the closing price at time t.

Analysis procedure

For analyzing the results, two subsections were created: one for portfolios that included transaction costs and another for portfolios that did not include these costs.

From this point, a comparison of the accumulated returns of the portfolios with respect to the benchmarks and volatility was initiated. Subsequently, the Sharpe ratio was analyzed, which is based on the returns of the portfolio, benchmark, and volatility as described by Sharpe (1966), thereby capturing the main points for investors in accordance with Markowitz (1952). The Sharpe ratio is calculated as shown in Eq. (9), and a higher value is preferable. (9) S=Rp−Rfσp

Wherein:

• S denotes the Sharpe Ratio; Rp is the expected return of the investment portfolio; Rf is the risk-free rate; and σp denotes the standard deviation of the investment portfolio.

To provide a more comprehensive perspective, the study conducted an analysis of the portfolios’ beta concerning the benchmarks. The beta, as introduced by Markowitz (1952), serves the purpose of assessing whether a specific asset or portfolio exhibits higher or lower volatility compared to the benchmark. The computation of beta, depicted by Eq. (10), indicates that a value below 1 signifies that the portfolio carries less risk than the market, while a value exceeding 1 suggests a higher risk profile. (10) β=CovRi,RmVarRm

Wherein:

• βi denotes the beta coefficient of asset i; Cov(ri, rm) represents the covariance between the returns of asset i and the market; and Var(rm) signifies the variance of market returns.

In summary, portfolios with higher returns, lower volatility, a higher Sharpe ratio, and a lower beta are preferable. It is important to note that a Sharpe ratio with a negative value should be interpreted cautiously, as its interpretation is likely misleading due to portfolios with higher volatility being considered superior when analyzing this index in isolation.

In this research, the methodology proposed by Ledoit & Wolf (2008) was employed to compare portfolios with different characteristics. The method uses the bootstrap approach to generate a distribution of the estimated difference in Sharpe ratios and construct a confidence interval to assess its statistical significance. This allows for the detection of significant differences between the compared portfolios.

The specific objective was to analyze the differences between the portfolios, considering transaction costs and the inclusion of commodities in the composition. The analysis sought to understand the impact of these factors on portfolio performance, providing valuable insights for efficient investment management, while benefiting from the robustness of the method used in detecting statistically significant differences between the Sharpe ratios of the compared portfolios.

Results

Portfolios without transaction costs

The performance of portfolios without transaction costs is detailed in Tables 2 and 3. The former exclusively encompasses equities, while the latter includes the incorporation of commodity derivatives. It is important to highlight that the PTF and MINVAR portfolios were constructed using traditional investment methodologies, specifically, Buy-and-Hold and Minimum Variance, respectively. In contrast, the other portfolios leveraged Artificial Intelligence techniques based on reinforcement learning, specifically the five RL algorithms previously mentioned. Notably, the best performance measures for the models are shown in Tables 2–5 by the bold numbers.

Table 2 Performance of equity portfolios without transaction costs and comparison with benchmark indices.

Rp denotes the average annual return (in percentage), Racum represents the accumulated return (in percentage), Vol indicates volatility, S provides the Sharpe ratio, β (IBOV) and β (DJI) are the portfolio betas with respect to the Ibovespa and Dow Jones indices, respectively. The best performance measures for the models are shown in bold.

	Benchmarks	Portfolios	
Metric	DJI	IBOV	SAC	DDPG	PPO	A2C	TD3	PTF	MINVAR	
R p	5.81	6.40	7.57	9.23	8.97	9.51	7.91	8.97	1.68	
R acum	28.70	31.33	29.53	36.75	35.60	37.97	30.96	35.59	6.08	
Vol	22.05	27.85	21.49	22.19	20.83	21.40	19.81	20.84	13.89	
S	0.367	0.364	0.448	0.510	0.518	0.532	0.484	0.518	0.190	
β (IBOV)	–	1.000	0.597	0.637	0.585	0.601	0.508	0.585	0.269	
β (DJI)	1.000	–	0.703	0.712	0.710	0.705	0.631	0.710	0.377	
Notes.

Source: research data.

Table 3 Performance of equity and commodity portfolios without transaction costs, and comparison with benchmark indices.

Rp is the average annual return (in percentage), Racum is the accumulated return (in percentage), Vol denotes volatility, S gives the Sharpe index, β (IBOV) and β (DJI) are the portfolio betas with respect to the Ibovespa and Dow Jones indices, respectively. The best performance measures for the models are shown in bold.

	Benchmarks	Portfolios	
Metric	DJI	IBOV	SAC	DDPG	PPO	A2C	TD3	PTF	MINVAR	
R p	5.81	6.40	12.00	11.58	10.68	11.07	10.73	10.68	5.91	
R acum	28.70	31.33	49.42	47.45	43.27	45.08	43.51	43.25	22.57	
Vol	22.05	27.85	17.92	18.17	17.95	18.78	17.91	17.95	9.62	
S	0.367	0.364	0.723	0.695	0.656	0.654	0.660	0.656	0.646	
β (IBOV)	–	1.000	0.495	0.510	0.501	0.531	0.485	0.501	0.139	
β (DJI)	1.000	–	0.600	0.613	0.603	0.644	0.590	0.602	0.194	
Notes.

Source: research data.

Table 4 Performance of stock portfolios including transaction costs and comparison with reference indices.

Rp denotes the average annual return (in percentage), Racum stands for the accumulated return (in percentage), Vol represents volatility, S supplies the Sharpe ratio, β (IBOV) and β (DJI) are the betas of the portfolios with respect to the Ibovespa and Dow Jones indices, respectively. The best performance measures for the models are shown in bold.

	Benchmarks	Portfolios	
Measure	DJI	IBOV	SAC	DDPG	PPO	A2C	TD3	PTF	MINVAR	
R p	5.81	6.40	7.64	8.43	8.98	8.04	11.56	8.97	1.74	
R acum	28.70	31.33	29.83	33.23	35.65	31.52	47.35	35.59	6.29	
Vol	22.05	27.85	20.44	21.16	20.82	21.12	20.69	20.84	13.89	
S	0.367	0.364	0.463	0.489	0.518	0.473	0.633	0.518	0.194	
β (IBOV)	–	1.000	0.554	0.598	0.585	0.600	0.578	0.585	0.269	
β (DJI)	1.000	–	0.670	0.724	0.710	0.702	0.698	0.710	0.377	
Notes.

Source: research data.

Table 5 Performance of stock and commodity portfolios, including transaction costs and comparison with reference indices.

Rp is the average annual return (in percentage), Racum is the accumulated return (in percentage), Vol represents volatility, S provides the Sharpe ratio, β (IBOV) and β (DJI) are the betas of the portfolios with respect to the Ibovespa and Dow Jones indices, respectively. The best performance measures for the models are shown in bold.

	Benchmarks	Portfolios	
Measure	DJI	IBOV	SAC	DDPG	PPO	A2C	TD3	PTF	MINVAR	
R p	5.81	6.40	9.14	12.45	10.67	10.77	11.25	10.68	5.91	
R acum	28.70	31.33	36.32	51.55	43.22	43.66	45.93	43.25	22.58	
Vol	22.05	27.85	18.23	18.12	17.93	17.44	17.49	17.95	9.62	
S	0.367	0.364	0.572	0.739	0.656	0.674	0.698	0.656	0.646	
β (IBOV)	–	1.000	0.508	0.498	0.500	0.486	0.489	0.501	0.139	
β (DJI)	1.000	–	0.601	0.588	0.602	0.580	0.590	0.602	0.194	
Notes.

Source: research data.

When comparing the returns of equity portfolios with the IBOV index, it becomes evident that the conventional MINVAR approach, as well as the SAC and TD3 portfolios, underperformed relative to the benchmark. Conversely, the remaining three AI portfolios outperformed the PTF portfolio in terms of returns. Upon the inclusion of commodities, only the MINVAR portfolio failed to surpass the reference index, whereas all five AI portfolios achieved returns exceeding those of traditional strategies.

Upon aligning the equity portfolio with the DJI, only MINVAR achieved a cumulative return lower than the index. However, the DDPG, PPO, and A2C portfolios once again outperformed traditional techniques. It is worth highlighting that, as indicated by the bold figures in Table 2, the A2C portfolio exhibited remarkable outperformance in terms of cumulative returns. With the inclusion of commodity derivatives in the analysis, only MINVAR yielded results below the benchmark, while the three superior portfolios continued to leverage AI techniques.

Furthermore, one can observe an approximate 12% increase in the average return of the portfolios when commodities are included, an outcome that surpasses both the IBOV and DJI by at least a 10% margin. This emphasises the import of including such assets, in addition to equities, in investment portfolios within volatile environments, corroborating the findings of Millea & Edalat (2022) and indicating a strong return as per Fama (1965). One may view these results in Figs. 1 and 2.

However, an investor should not focus solely on achieving a robust return but should also aim for low volatility to mitigate risk (Markowitz, 1952). The results obtained from the equity investment portfolios, except for the traditional MINVAR technique, exhibited volatility ranging from 19.81% (TD3) to 22.19% (DDPG), a factor potentially associated with the lower returns observed. However, upon the inclusion of commodities, the volatility of all portfolios decreased, emphasizing the significance of incorporating these assets and further supporting the findings of Millea & Edalat (2022). Interestingly, when compared to the benchmarks, none of the equity portfolios displayed higher volatility than the IBOV, and only one portfolio (DDPG) exhibited higher volatility than the DJI. When commodities were taken into account, none of the portfolios exceeded the benchmark in terms of volatility.

When evaluating the Sharpe Index, it becomes evident that the top-performing portfolios consisting solely of equities were A2C, PTF, and PPO, with results differing by less than 0.02 among them. It is apparent that the Buy-and-Hold (B&H) strategy entailed fewer risks than some AI-based portfolios. Conversely, with the addition of commodities to the portfolios, the best-performing choices were SAC, DDPG, and TD3, with differences among them being less than 0.07, showing an increase of approximately 0.2 compared to those containing only equities. Moreover, only the MINVAR portfolio yielded results inferior to the IBOV and DJI, and no portfolio with the addition of commodities underperformed these benchmarks.

In the assessment of the Beta Index, it is evident that none of the portfolios exhibit a higher level of risk compared to the benchmark indices. Notably, beta values decrease with the inclusion of commodity derivatives, reinforcing the consistent pattern of improved results when these assets are integrated. In all cases, the MINVAR portfolio consistently demonstrated the lowest beta value, concurrently exhibiting the weakest performance in terms of returns. This underscores the importance of a more comprehensive analysis, as exemplified by the Sharpe Index. Consequently, it becomes evident that, overall, the portfolios outperformed both the MINVAR and PTF portfolios.

This observation challenges the validity of the efficient market hypothesis, suggesting that the model proposed by Lo (2004) may offer a more suitable framework. The superior performance exhibited by portfolios that integrate technical indicators as influential variables in their evolution lends substantial credence to this perspective.

Figure 1 Accumulated return of share portfolios without transaction costs.

Source: Authors.

Figure 2 Accumulated return of share portfolios and commodity derivatives, with no transaction costs.

Source: Authors.

Portfolios incurring transaction costs

This section addresses the concern regarding the potential costliness of portfolio management when utilizing computational tools, particularly in terms of transaction costs. As demonstrated by the performance indicators outlined in Tables 4 and 5, as well as the visual analysis of portfolio progression depicted in Figs. 3 and 4, it becomes apparent that the inclusion of transaction costs leads to variations in portfolio return outcomes. Notably, these outcomes reveal that returns can even surpass those achieved without transaction costs. This phenomenon arises from the algorithm’s ability to make fewer transactions or opt for different asset acquisitions.

Figure 3 Accumulated return of the share portfolios with fees.

Source: Authors.

Figure 4 Accumulated return of share and commodities portfolios with fees.

Source: Authors.

The results presented in Tables 4 and 5 unequivocally indicate that two equity portfolios generated cumulative returns lower than the IBOV (SAC and MINVAR), with only one falling short in comparison to the DJI (MINVAR). However, with the inclusion of commodities in the portfolio, exclusively the MINVAR portfolio exhibited returns below both benchmark indices. In terms of equity portfolios, the top three performers, in sequence, were TD3, PPO, and PTF. With the incorporation of commodities, these rankings shifted to DDPG, TD3, and A2C. Consequently, the returns from AI-driven portfolios once again demonstrate a propensity to outperform those derived from conventional techniques.

Additionally, it is essential to highlight that the mean returns exhibit a notable increase of 9% when commodities are incorporated. In the scenario where only equities are utilized, both mean returns surpass those of the benchmarks. However, upon the inclusion of commodities, the mean returns experience an increase of at least 9% compared to both benchmarks. It is worth emphasizing that the average results remain relatively consistent when comparing portfolios that consider transaction costs with those that do not. The equity portfolio demonstrates a 1% superiority with transaction costs, and a marginally lower performance of nearly 2% when commodities are included.

When examining the volatility of the share portfolios, it is seen that the outcomes are rather similar, with yields approximately 20% per annum, with the exception of the MINVAR portfolio, which falls notably below this average. When commodities are included, the outcomes of all the portfolios also drop. In a manner similar to the portfolios that do not consider transaction costs, none of them achieved a volatility superior to that of the IBOV, nor in relation to the DJI, considering the portfolios with or without commodities.

When analyzing the Sharpe ratio results, it becomes clear that the three top-performing equity portfolios are TD3, PPO, and PTF, in that respective order. However, the differences between them are more pronounced compared to scenarios without transaction costs, with an increase exceeding 0.1. With the inclusion of commodities, we observe that the best-performing portfolios are DDPG, TD3, and A2C, with differences among them amounting to 0.06. While these variations differ in magnitude when compared to scenarios without commodities, there is a consistent overall increase in Sharpe indices, albeit of a lesser scale than in scenarios without transaction costs. Notably, among the equity portfolios, only the MINVAR portfolio yielded returns below those of the IBOV and DJI, and none performed worse when commodities were included.

In the portfolios considering transaction costs, as occurred in those that did not consider them, it is observed that the β index is less than 1 for all cases, and the inclusion of commodities results in a reduction of all their outcomes.

Lastly, it is worth emphasizing that, as with the portfolios that did not take into account transaction costs, the portfolios, in general, presented superior results to MINVAR, which is based on the minimum-variance technique, and some outcomes were superior to PTF, which is based on the B&H strategy. It is vital to consider the relevance of these outcomes for the development of more effective investment strategies.

Statistical tests

To ascertain the statistical superiority of portfolio performances, we employed the Ledoit-Wolf Test (LW Test) developed by Ledoit & Wolf (2008) to analyze variations in Sharpe Ratios. Our analysis consisted of several comparisons to shed light on the performance of various strategies under different portfolio scenarios. The outcomes are summarized in Tables 6 and 7.

Regardless of the particular case, our investigation showed that the MINVAR method consistently underperformed the other techniques. In the portfolio that included commodities but excluded transaction expenses, TD3 showed a statistically significant lower Sharpe ratio than A2C, DDPG, and SAC. However, despite the fact that A2C appeared to perform better than the others, there was no discernible difference between DDPG, SAC, or A2C.

To further scrutinize the disparities in Sharpe ratios among all models in different investment scenarios, we conducted the LW Test across three key comparisons: “Commodities” evaluates the impact of including this asset class in portfolios; “Transaction Costs” which examines the effects of taking into account transaction costs; and “Both”, that combines both factors. These results are detailed in Table 8.

Table 6 Z-score statistics from LW Test comparing Sharpe ratios in portfolios with commodities (above diagonal) and portfolios without commodities (below diagonal).

	A2C	DDPG	MINVAR	PPO	PTF	SAC	TD3	
A2C	–	0,75	14,866★	1,13	1,132	1,051	1,171	
DDPG	−0,762	–	14,536★	0,399	0,401	0,316	0,442	
MINVAR	9,292★	9,707★	–	−31,218★	−31,219★	−31,169★	−31,244★	
PPO	0,626	1,323	−27,388★	–	0,002	−0,085	0,044	
PTF	0,624	1,32	−27,405★	−0,002	–	−0,088	0,042	
SAC	0,065	0,787	−32,111★	−0,586	−0,584	–	0,129	
TD3	1,764•	2,411⧫	−20,547★	1,184	1,186	1,706•	–	
Notes.

Significance levels of 1%, 5%, and 10% are represented by ★,⧫, and•, respectively.

Table 7 Z-score statistics from LW Test (Ledoit & Wolf, 2008) comparing Sharpe ratios in portfolios with transaction costs (above diagonal) and portfolios without costs (below diagonal).

	A2C	DDPG	MINVAR	PPO	PTF	SAC	TD3	
A2C	–	−0,081	9,054★	0,228	0,225	0,737	0,163	
DDPG	−1,186	–	9,100★	0,308	0,305	0,814	0,242	
MINVAR	14,048★	14,546★	–	−27,321★	−27,343★	−24,015★	−27,796★	
PPO	−0,73	0,422	−31,217★	–	−0,003	0,517	−0,067	
PTF	−0,744	0,409	−31,21★	−0,013	–	0,52	−0,064	
SAC	−1,016	0,157	−31,066★	−0,273	−0,259	–	−0,604	
TD3	−0,124	0,983	−31,598★	0,577	0,59	0,835	–	
Notes.

Significance levels of 1%, 5%, and 10% are represented by ★,⧫, and•, respectively.

Table 8 Z-score statistics of the Ledoit & Wolf (2008) test (LW Test) assessing the variance in Sharpe ratios across different model applications in three scenarios: portfolios with and without commodities, with and without transaction costs, and combining both factors.

	LW Test for each model	
Comparison	A2C	DDPG	MINVAR	PPO	PTF	SAC	TD3	
Commodities	−2,680★	−3,741★	−71,231★	−3,863★	−3,870★	−3,086★	−4,008★	
Transaction Costs	0,405	1,036	−0,009	0,001	–	1,063	−1,358	
Both	3,825★	2,977★	10,212★	3,073★	3,065★	2,409⧫,	3,600★	
Notes.

Significance levels of 1%, and 5% are represented by ★, and⧫, respectively.

According to our investigation, all models and the combined scenario showed statistically considerably better results for portfolios that included commodities. The impact of transaction costs on portfolio performance, however, did not show any statistically significant changes.

The robustness of these results suggests that certain models excel in specific contexts, offering useful evidences for both investors and academics. Understanding which models perform optimally when considering factors as the inclusion of commodities and transaction costs can guide more effective portfolio management. Thus, these findings can be deemed not only interesting but also relevant to the fields of investment and finance.

Portfolio composition analysis

The examination into asset weight distribution, classified by countries or commodities, uncovers intriguing variations across the models and portfolio types under scrutiny. Detailed outputs can be found in Table 9.

Table 9 Descriptive statistics of asset class weights in portfolios excluding commodities.

	No transaction costs	Transaction Costs Included	
Asset Class	A2C	DDPG	PPO	SAC	TD3	A2C	DDPG	PPO	SAC	TD3	
Portfolio without Commodities	
Minimum weights	
Brazil	0.25	0.25	0.25	0.228	0.194	0.25	0.25	0.25	0.227	0.25	
USA	0.202	0.138	0.25	0.188	0.194	0.191	0.25	0.25	0.204	0.25	
Europe	0.25	0.25	0.25	0.25	0.25	0.25	0.244	0.249	0.25	0.199	
China	0.25	0.25	0.25	0.25	0.25	0.237	0.204	0.25	0.25	0.25	
Maximum weights	
Brazil	0.28	0.335	0.25	0.278	0.25	0.295	0.28	0.25	0.25	0.273	
USA	0.25	0.25	0.25	0.25	0.25	0.25	0.28	0.251	0.25	0.267	
Europe	0.255	0.286	0.25	0.295	0.293	0.28	0.256	0.25	0.286	0.25	
China	0.265	0.269	0.25	0.286	0.318	0.25	0.25	0.25	0.303	0.273	
Average Weights	
Brazil	0.278	0.321	0.25	0.256	0.194	0.293	0.259	0.25	0.229	0.269	
USA	0.206	0.154	0.25	0.198	0.194	0.192	0.27	0.251	0.206	0,26	
Europe	0.253	0.267	0.25	0.285	0.293	0.278	0.249	0.249	0.275	0.201	
China	0.263	0.258	0.25	0.261	0.318	0.237	0.222	0.25	0.29	0.269	
Standard Deviation of Weights	
Brazil	0.002	0.008	0	0.013	0.002	0.002	0.01	0	0.003	0.003	
USA	0.002	0.01	0	0.004	0.002	0.002	0.007	0	0.003	0.008	
Europe	0.001	0.014	0	0.005	0.001	0.001	0.003	0	0.004	0.003	
China	0.001	0.004	0	0.01	0.002	0	0.008	0	0.008	0.003	
Portfolio with Commodities	
Minimum weights	
Brazil	0.2	0.2	0.2	0.185	0.166	0.2	0.2	0.2	0.2	0.185	
USA	0.2	0.2	0.2	0.188	0.196	0.192	0.186	0.2	0.183	0.169	
Europe	0.194	0.164	0.2	0.2	0.172	0.18	0.152	0.2	0.153	0.2	
China	0.17	0.194	0.2	0.15	0.182	0.192	0.2	0.2	0.192	0.182	
Commodities	0.2	0.187	0.2	0.2	0.2	0.2	0.2	0.2	0.2	0.2	
Maximum weights	
Brazil	0.211	0.219	0.2	0.2	0.2	0.21	0.211	0.2	0.236	0.208	
USA	0.222	0.228	0.2	0.212	0.223	0.2	0.2	0.2	0.2	0.208	
Europe	0.2	0.2	0.2	0.231	0.2	0.2	0.2	0.2	0.2	0.223	
China	0.2	0.203	0.2	0.2	0.204	0.2	0.238	0.2	0.204	0.2	
Commodities	0.203	0.212	0.2	0.227	0.251	0.234	0.229	0.2	0.246	0.227	
Average Weights	
Brazil	0.21	0.212	0.2	0.192	0.182	0.204	0.208	0.2	0.211	0.192	
USA	0.222	0.222	0.2	0.2	0.21	0.193	0.19	0.2	0.192	0.188	
Europe	0.194	0.166	0.2	0.218	0.178	0.181	0.159	0.2	0.158	0.214	
China	0.171	0.197	0.2	0.179	0.196	0.192	0.217	0.2	0.199	0.185	
Commodities	0.203	0.203	0.2	0.212	0.234	0.229	0.226	0.2	0.239	0.221	
Standard Deviation of Weights	
Brazil	0.001	0.004	0	0.003	0.008	0.005	0.002	0	0.014	0.008	
USA	0.001	0.004	0	0.009	0.008	0.001	0.002	0	0.007	0.009	
Europe	0	0.003	0	0.007	0.002	0.001	0.007	0	0.004	0.009	
China	0.001	0.003	0	0.009	0.004	0	0.008	0	0.005	0.002	
Commodities	0	0.007	0	0.003	0.014	0.005	0.002	0	0.006	0.006	

Specifically, when examining portfolios that exclude both transaction costs and commodities (Table 9), significant disparities in investment allocations to China and Europe emerge among different models. Notably, the TD3 model stands out with a preference for China, reflecting a weight of 0.32. In portfolios that incorporate commodities but do not account for transaction costs, all models–except SAC–display a nearly uniform distribution among asset classes. However, when transaction costs are included, TD3 assumes a more substantial position in Brazil and the USA.

In portfolios that comprise both stocks and commodities without considering transaction costs, the TD3 model exhibits a notable inclination towards commodities. This preference is evident in its highest maximum, minimum, and average weights (25%, 20%, and 23.4%, respectively) and low standard deviation. In contrast, the other models maintain a more balanced allocation among asset classes.

Regarding portfolios exclusively composed of stocks, most models maintain nearly equal distribution among various countries. However, SAC and TD3 display a slight preference for the European and Chinese markets, respectively. These variations in weight allocations signify the unique strategies and learning patterns employed by each model, often shaped by diverse hyperparameter configurations. This highlights the significance of considering multiple approaches when constructing diversified investment portfolios.

Moreover, Table 9 provides extra and intriguing data that offers implicit deductions about these findings. As supplementary information, the complete historical data, including weight distribution, is accessible in the data repository, providing a comprehensive resource for further analysis.

Conclusions

This research probed the efficacy of reinforcement learning (RL) algorithms in the optimization of investment portfolios, drawing comparisons with conventional strategies and examining the repercussions of incorporating commodities as well as the effect of transaction costs.

The results suggest that, in general, artificial intelligence (AI) techniques surpassed traditional methods in performance. Furthermore, the incorporation of commodities significantly contributed to improving portfolio performance while mitigating volatility. Although favorable returns observed when accounting for transaction costs, they did not exert significant impact on the results. Yet, the distribution of asset weights varies significantly among reinforcement learning models, reflecting distinct strategies, potential regional preferences, and sensitivity to transaction costs, underscoring the importance of diversified approaches to portfolio construction.

This study makes a significant contribution to the academic literature by introducing an innovative methodology for optimizing investment portfolios and providing interesting findings into the application of AI techniques in this domain. Additionally, this research highlights the importance of asset diversification, including commodities, and analyzes the impact of transaction costs on model learning, thereby expanding our understanding of the possibilities and challenges associated with this state-of-the-art approach.

Nonetheless, this study is not without limitations. For instance, it relies solely upon two benchmark indices (Dow Jones and Ibovespa), which may circumscribe the generalization of results to other financial markets. Additionally, the deployment of a constrained dataset of historical data for RL model training might impinge upon the algorithm’s ability to forecast future events and make real-time strategy adjustments. Another constraining factor is the omission of other pertinent factors in asset selection, such as fundamental and technical analyses. Moreover, the study did not contemplate the ramifications of exogenous events such as governmental policy alterations, economic shifts, extreme weather phenomena, wars, and others, which can substantially affect the performance of financial assets. Lastly, the investigation encompassed only a limited repertoire of RL algorithms and conventional investment strategies, which might further limit the generalization of the findings.

Future research avenues could encompass performance analyses of RL algorithms across various financial market epochs, to glean insights into these models’ adaptability to scenario shifts. Furthermore, extending the scope of analysis regarding the inclusion of commodities in investment portfolios to different asset classes and contrasting their effects on other performance metrics beyond the Sharpe Index and beta would be valuable. A captivating area to explore entails the integration of ethical and social criteria in portfolio optimization through AI algorithms, taking into consideration factors such as sustainability and social responsibility. Lastly, examining avenues to mitigate the effects of transaction costs on portfolio optimization through AI algorithms could enhance these models’ adaptability across diverse market conditions.

Additional Information and Declarations

Competing Interests

Author Contributions

Data Availability

The authors declare there are no competing interests. Gustavo Carvalho Santos is one of the cofounders of Sapiens Agro (Sapiens Inteligênciaa Ltda).

Gustavo Carvalho Santos conceived and designed the experiments, performed the experiments, analyzed the data, performed the computation work, prepared figures and/or tables, authored or reviewed drafts of the article, and approved the final draft.

Daniel Garruti conceived and designed the experiments, analyzed the data, prepared figures and/or tables, authored or reviewed drafts of the article, and approved the final draft.

Flavio Barboza conceived and designed the experiments, performed the experiments, analyzed the data, performed the computation work, prepared figures and/or tables, authored or reviewed drafts of the article, and approved the final draft.

Kamyr Gomes de Souza conceived and designed the experiments, analyzed the data, prepared figures and/or tables, authored or reviewed drafts of the article, and approved the final draft.

Jean Carlos Domingos conceived and designed the experiments, performed the experiments, analyzed the data, performed the computation work, prepared figures and/or tables, authored or reviewed drafts of the article, and approved the final draft.

Antônio Veiga conceived and designed the experiments, analyzed the data, authored or reviewed drafts of the article, and approved the final draft.

The following information was supplied regarding data availability:

The code and raw data are available at Harvard Dataverse: Barboza, Flavio, 2023, ”Replication Data and Code for: Management of Investment Portfolios Employing Reinforcement Learning”, https://doi.org/10.7910/DVN/JEKGJ8, Harvard Dataverse, V4, UNF:6:pzIxOUBN9tI9/z6n71pDHQ== [fileUNF].

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
