# Peer review of "Management of investment portfolios employing reinforcement learning"

_PeerJ Computer Science, doi:10.7717/peerj-cs.1695_

## Round 0.1 · original submission · Major Revisions

The reviewers, who are experts in portfolio optimization, are rather positive about the paper, but they require significant improvements (in particular Rev #1). I agree with them and I encourage the authors to improve the paper along the lines suggested in the reviews and resubmit it.

**Language Note:** The review process has identified that the English language must be improved. PeerJ can provide language editing services - please contact us at [email protected] for pricing (be sure to provide your manuscript number and title). Alternatively, you should make your own arrangements to improve the language quality and provide details in your response letter. – PeerJ Staff

Reviewer 1 ·

Basic reporting

Management of investment portfolios employing reinforcement learning

(#89002)
Section 1: Introduction and general comments
In this contribution, the authors address the effectiveness of Reinforcement Learning (RL) algorithms in the optimisation of investment portfolios and compare them with conventional strategies, as well as with the American and Brazilian indices. Additionally, the authors assess the costs and benefits of incorporating commodities derivatives into portfolios. The findings denote that the inclusion of derivatives can markedly enhance portfolio performance whilst attenuating volatility, thus emerging as an attractive alternative for investors.
The authors present a comprehensive analysis of RL-based portfolio optimization; however, the manuscript requires extensive editing. I have some major concerns w.r.t. the discussion of the methodology, which is not sufficiently accurate. I also expect the authors to further elaborate on the results. A major revision is therefore recommended.

Section 2: Specific comments

Comment #1: English
A number of expressions in English are slightly weird, resulting often in unclear statements. A selection is reported below, for which rephrasing/reformulation in good English is required. A more in-depth editing is left to the authors. Altogether, the paper would also benefit from thorough editing and proofreading, also for grammar, spelling and punctuation correction.

A few examples:

-Line 15-16, page 1: “Nevertheless, this endeavour can prove to be formidable, costly, and intricate”.
-Line 18-19, page 1: “Additionally, we evaluate ramifications by incorporating commodities derivatives into portfolios and the transaction costs associated therewith”.
-Line 23-24, page 1: “[…] by elucidating the application of RL and imparting implications for academia”.
-Line 26-27, page 1: “[…] while concurrently mitigating conflicts of interest and obviating biased decision-making” (in this case the statement is pretty much unclear, a reasoned rearrangement is required).
-Line 47-48, page 2: “This research endeavour aims to investigate the efficacy of RL algorithms in the optimisation of investment portfolios”.
-Line 64, page 2: “In terms of outcomes, it has been observed that when analysing portfolios sans transaction costs”;
-Line 164-165, page 4: “DRL has achieved remarkable triumphs across a wide spectrum of
intricate tasks, encompassing gaming, robotics, and system optimisation”.
-Line 227-228, page 4: “In conclusion, Song et al. (2022) proffered a novel approach for optimising investment portfolios, employing stochastic reinforcement learning”.
-Line 238-239, page 4: “Yahoo Finance (2022) furnished the historical data for the equities and usually applied in similar studies (Xiao and Ihnaini, 2023), while the 240 commodity data stemmed from Barchart (2022)”.
-Line 243-244, page 5: As elucidated by Markowitz (1952), diversification remains a paramount strategy for risk management and portfolio optimisation.
-Line 460-461, page 11: “This section confronts the suspicion that portfolio management through computational tools is costly in terms of transaction costs”.
-Replace in Tables 1-4 “References” with Benchmarks.

Experimental design

Comment #2: Methodology

-Reproducibility and replicability are important for ensuring the quality and integrity of scientific research and to guarantee accurate and trustworthy results, that are applicable to a wider range of datasets, as the authors suggest in the conclusions. However, the discussion of methods resembles more a review of RL strategies, rather than a proper presentation. Therefore, I recommend the authors to fix the section, which should focus in particular on a formal discussion of the models.
This is a major flaw since it makes also difficult to assess the parameters involved and what might explain possible differences in performance

Validity of the findings

Comment #3: Results

3.1-Regarding the out-of-sample results presented in Section “Results”, it is important to assess if the reported differences are statistically significant or not. At least I recommend to report the bootstrap p-values for the difference between Sharpe ratios and for the difference between variances. This can be done using the robust methodology proposed by Ledoit and Wolf (2008).

Reference:
O. Ledoit and M. Wolf (2008). "Robust performance hypothesis testing with the Sharpe ratio". Journal of Empirical Finance, Vol. 15, No. 5, pp. 850-859.


3.2-I would like to see a more in-depth analysis of the results. The authors include commodity futures in the portfolio and claim that such inclusion is beneficial for portfolio risk returns profiles. I suggest, for instance, to assess and to show how weights evolve over time, maybe aggregating all the assets by classes for better visualization.

Additional comments

no comments

Cite this review as

Reviewer 2 ·

Basic reporting

- The paper is clearly written. The use of English is acceptable.
- Acceptable background provided. Authors are encouraged to review the latest survey paper “Portfolio Optimisation Problem: A Taxonomic Review of Solution Methodologies”.
- Should list contributions but not report findings and conclusion in the Introduction section.

Experimental design

- This research work is within Aims and Scope of the journal.
- Research question is well defined.
- Methods were described but with insufficient details for replication.

Validity of the findings

- The RL algorithms were tested on limited benchmark.

Additional comments

- This paper investigate the efficacy of RL algorithms compared to traditional strategies, in the optimisation of investment portfolios (Dow Jones and Ibovespa indices as benchmarks), employing the FinRL library. The findings indicate that, on the whole, artificial intelligence (AI) techniques outperformed traditional methods. Moreover, the inclusion of commodities proved instrumental in enhancing portfolio performance substantially while curtailing volatility. Transaction costs wielded considerable influence over the outcomes, with more favourable returns achieved when accommodating these costs.
- This paper contributes little technical contributions in terms of methodology.
- Redundant paragraphs were detected in the Methods section (Lines 294-327).
- Inconsistent use of decimal points (dots for Tables 1 and 2 and commas for Tables 3 and 4).
- There is no mention of the meaning of bolded values in Table 1.
- The organization of the paper can be improved. For example, in methods section, one paragraph should be used to explain each RL instead of two, to improve clarity.

Cite this review as

---

## Round 0.2 · accepted · Accept

Both reviewers believe that the authors have fulfilled their requests in a satisfactory way. I looked at the paper and the answers, and I agree with them.

Reviewer 1 ·

Basic reporting

Please refer to my previous comments for a description of the paper.
The authors answered all my propositions and suggestions in a satisfactory manner.
I therefore suggest to accept the paper.

Experimental design

see sect.1

Validity of the findings

see sect.1

Additional comments

see sect.1

Cite this review as

Reviewer 2 ·

Basic reporting

The authors have addressed most of all the comments raised previously.

Experimental design

The authors have addressed most of all the comments raised previously.

Validity of the findings

The authors have addressed most of all the comments raised previously.

Cite this review as